# Poly(ADP-ribose)-binding protein RCD1 is a plant PARylation reader regulated by Photoregulatory Protein Kinases

Julia P. Vainonen[1,9], Richard Gossens[1,9], Julia Krasensky-Wrzaczek[1,2,9], Raffaella De Masi[3,4], Iulia Danciu[5,6], Tuomas Puukko[1], Natalia Battchikova[7], Claudia Jonak [5,6], Lennart Wirthmueller[3,4], Michael Wrzaczek[1,2], Alexey Shapiguzov[1,8] & Jaakko Kangasjärvi [1✉]

Poly(ADP-ribosyl)ation (PARylation) is a reversible post-translational protein modification that has profound regulatory functions in metabolism, development and immunity, and is conserved throughout the eukaryotic lineage. Contrary to metazoa, many components and mechanistic details of PARylation have remained unidentified in plants. Here we present the transcriptional co-regulator RADICAL-INDUCED CELL DEATH1 (RCD1) as a plant PAR-reader. RCD1 is a multidomain protein with intrinsically disordered regions (IDRs) separating its domains. We have reported earlier that RCD1 regulates plant development and stress-tolerance by interacting with numerous transcription factors (TFs) through its C-terminal RST domain. This study suggests that the N-terminal WWE and PARP-like domains, as well as the connecting IDR play an important regulatory role for RCD1 function. We show that RCD1 binds PAR in vitro via its WWE domain and that PAR-binding determines RCD1 localization to nuclear bodies (NBs) in vivo. Additionally, we found that RCD1 function and stability is controlled by Photoregulatory Protein Kinases (PPKs). PPKs localize with RCD1 in NBs and phosphorylate RCD1 at multiple sites affecting its stability. This work proposes a mechanism for negative transcriptional regulation in plants, in which RCD1 localizes to NBs, binds TFs with its RST domain and is degraded after phosphorylation by PPKs.

[1] Organismal and Evolutionary Biology Research Programme, Faculty of Biological and Environmental Sciences, and Viikki Plant Science Center, University of Helsinki, FI-00014 Helsinki, Finland. [2] Institute of Plant Molecular Biology, Biology Centre, Czech Academy of Sciences, Branišovská1160/31, 370 05 České Budějovice, Czech Republic. [3] Department Biochemistry of Plant Interactions, Leibniz Institute of Plant Biochemistry, Weinberg 3, 06120 Halle (Saale), Germany. [4] Dahlem Centre of Plant Sciences, Institute of Biology, Freie Universität Berlin, Königin-Luise-Str. 12-16, 14195 Berlin, Germany. [5] Gregor Mendel Institute, Austrian Academy of Sciences, Vienna BioCenter, Dr. Bohr-Gasse 3, 1030 Vienna, Austria. [6] Bioresources Unit, Center for Health & Bioresources, AIT Austrian Institute of Technology GmbH, Konrad Lorenz Straße 24, 3430 Tulln, Austria. [7] Department of Biochemistry, Molecular Plant Biology, University of Turku, FI-20014 Turku, Finland. [8] Natural Resources Institute Finland (Luke), Production Systems, Toivonlinnantie 518, FI-21500 Piikkiö, Finland. [9] These authors contributed equally: Julia P. Vainonen, Richard Gossens, Julia Krasensky-Wrzaczek. ✉email: jaakko.kangasjarvi@helsinki.fi

Plant survival in a changing environment requires continuous reprogramming of gene expression in response to external signals. Signaling through heavily-regulated hub proteins that interact with many different protein partners provides a sophisticated system to adjust cellular functions according to the surrounding environment[1–3]. The *Arabidopsis thaliana* RADICAL-INDUCED CELL DEATH1 (RCD1) is a plant-specific protein proposed to function as a hub protein that acts as a negative transcriptional co-regulator of numerous stress and developmental responses in plants[1,3–5]. Together with its paralog *SIMILAR TO RCD ONE 1* (*SRO1*), which was formed in a partial genome duplication specific to Brassicaceae, RCD1 is essential for e.g., proper embryo development in Arabidopsis – the *rcd1 sro1* double mutant is lethal[4]. RCD1 interacts with over 30 transcription factors (TFs) through its C-terminal RCD1-SRO-TAF4 (RST) domain[3–6]. However, the regulation of RCD1 activity and the function of its other domains has remained an open question.

The *SRO* gene family[4,7,8], which encodes two protein groups with different domain architecture, is conserved in all land plants[8]. The type-A SROs (RCD1 and SRO1 in Arabidopsis; Fig. 1a) are multidomain proteins with intrinsically disordered regions (IDRs). They contain two N-terminal Nuclear Localization Sequences (NLS), a WWE domain, a Poly(ADP-ribose) polymerase-like (PARP-like) domain, and the C-terminal RST domain[4,7,8] separated by four IDRs. The type-B SROs (SRO2-SRO5 in Arabidopsis) lack the NLSs and the WWE domain[8]. The WWE domain was originally proposed to be a protein-protein interaction domain in proteins related to ubiquitination and ADP-ribosylation[9]. Later studies have, however, shown that some, but not all animal WWE domains bind *iso*-ADP ribose, a structural unit of poly(ADP-ribose) (PAR)[10,11]. In the Arabidopsis proteome, the WWE domain has been identified with high confidence only in RCD1 and its paralog SRO1. While the RCD1 and SRO1 proteins do not exhibit detectable PARP activity[8,12], or mono(ADP-ribosyl) transferase (MART) activity[13], the presence of the WWE and PARP-like domains together suggests a function of RCD1 in PAR-related processes[14]. Furthermore, the Arabidopsis SRO2 has demonstrated MART activity, which counteracts ubiquitination for protein homeostasis in plant immunity responses[13]. This suggests that the SRO proteins could also be mechanistically connected to ADP-ribosylation-mediated control of protein stability.

Poly(ADP-ribosyl)ation (PARylation) and mono(ADP-ribosyl)ation (MARylation) of proteins are dynamic and transient post-translational modifications that play critical roles in the adjustment of development and response to various stress conditions[15,16]. PARylation is catalyzed by PARPs, which covalently attach ADP-ribose moieties to specific amino acid residues in a species- and tissue-specific manner[17–21]. MARylation is catalyzed by MARTs, which attach a single ADP-ribose moiety to the target protein, but its role is less understood. PAR-glycohydrolase (PARG) can trim down PAR chains to the terminal protein-bound ADP-ribose thereby removing protein PARylation. Several signaling components that recognize PARylated proteins, PAR-readers, have been identified in animal systems[15,16]. In plants, however, PAR-readers have not been described yet.

On a functional level, PARylation has been shown to regulate a variety of cellular processes in animal cells, including chromatin remodeling, transcription, and programmed cell death[15,16]. Defects in PARylation have been associated with numerous metabolic, developmental, and neuronal diseases in humans. Consequently, this modification has been extensively studied in the animal field during the recent decades[15,22], and hundreds of PARylated proteins have been identified[17–19]. In plants, the role of PAR is only starting to emerge and, mostly due to methodological limitations, only eight proteins have unambiguously been shown to become PARylated[13,23–28]. Moreover, the consequences of protein PARylation, and especially how plants de-code PARylation events have remained almost completely unknown.

Besides interaction with transcription factors through the RST domain, RCD1 has been shown to interact in a non-RST-dependent manner with Photoregulatory Protein Kinases[5,12] (PPKs; also named MUT9-like kinases, MLKs, or Arabidopsis EL1-like kinases, AELs). In Arabidopsis, this recently discovered family of protein kinases is comprised of four members that localize to nuclear bodies (NBs), subnuclear non-membrane bound complexes of mostly unknown function[29]. PPKs interact with different nuclear proteins, including histones, components of the circadian clock and light signaling, and the ABA receptor PYR/PYL/RCAR[30–36]. PPK-dependent phosphorylation has been shown to target proteins for degradation[32,33,35], thereby playing an essential role in protein turnover.

Here we show that RCD1 directly binds PAR via its N-terminal WWE domain in-vitro and that PAR binding determines the subnuclear localization of RCD1 in nuclear bodies (NBs). PPKs co-localize with RCD1 in NBs and affect RCD1 function and stability by phosphorylating residues of the IDR between the WWE and PARP-like domains. Our results show that RCD1 represents a potential plant equivalent of mammalian PAR-reader proteins and provide insights into the mechanisms of how RCD1 activity is regulated in the cell.

## Results

**Each domain of the multidomain protein RCD1 is essential for its function**. In the SRO protein family, Arabidopsis RCD1 is the best-characterized member. Loss of functional RCD1 causes a wide range of plant phenotypes, growth/developmental disorders, and misregulated stress responses. In all *rcd1* alleles the phenotype is caused by lack of interaction with transcription factors since all alleles that display phenotypes are premature stop codons resulting in proteins missing the RST domain. The importance of the NLS, WWE and PARP-like domains in RCD1 has so far remained unstudied. To elucidate the function of RCD1 beyond TF binding, we generated RCD1 domain deletion variants (Supplementary Fig. 1a), re-introduced them into an *rcd1* null-mutant and analyzed the plants for complementation of *rcd1* phenotypes in plant habitus, tolerance to methyl viologen (MV), abundance of mitochondrial alternative oxidase (AOX1/2) protein, and flowering time. Our experiments showed that all these domains of RCD1 are essential for its function. Only the wild-type RCD1 construct fully reverted the leaf shape phenotype and the early flowering specific for *rcd1* (Fig. 1b and Supplementary Fig. 1c). In addition to retaining the *rcd1* growth phenotypes, the lines expressing the RCD1 deletion variants also displayed the *rcd1*-specific high abundance of the AOX1/2 proteins (Supplementary Fig. 1d) and MV tolerance (Fig. 1c and Supplementary Fig. 1b).

**Subnuclear localization into NBs is PAR-dependent**. RCD1 is targeted to the nucleus by a bipartite NLS located N-terminally of the WWE domain. Using confocal microscopy, we confirmed the function of the NLS in stable transgenic lines expressing Venus-tagged wild-type RCD1 or RCD1*nls* mutants; without the two NLSs the RCD1-Venus fusion protein did not enter the nucleus (Supplementary Fig. 2a). Phenotype analyses of wild-type RCD1-HA and RCD1*nls*-HA lines revealed that nuclear localization of RCD1 was essential for its function: despite increased protein levels of RCD1*nls*-HA (Supplementary Fig. 2b), expression of this variant in *rcd1* displayed the mutant-type plant habitus and leaf shape (Fig. 1d) and tolerance to MV (Fig. 1e). Within the nucleus, RCD1-Venus localized to the nucleoplasm and, intriguingly, to distinct NBs (Fig. 2a). Deletion of the WWE or the PARP-like domain, but not the RST domain, prevented the localization of

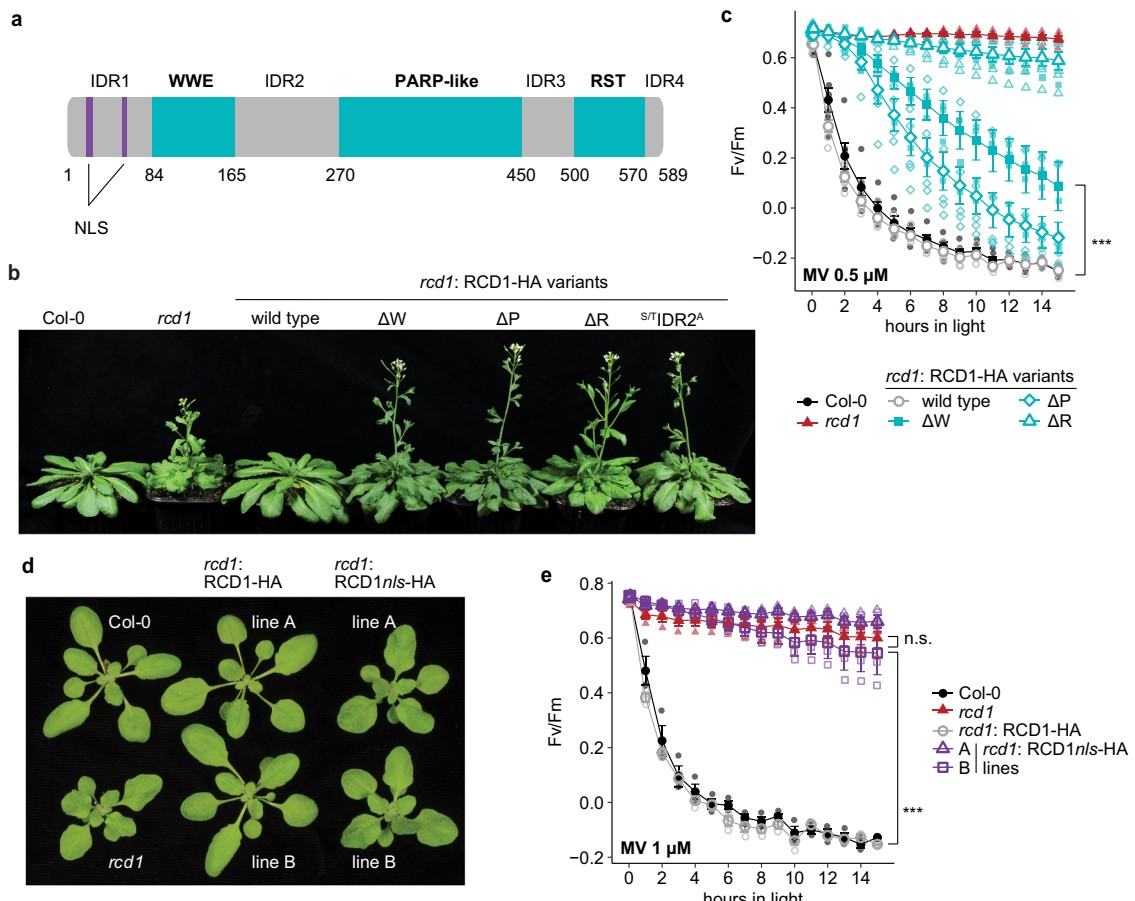

**Fig. 1 Nuclear localization and all three domains of RCD1 are essential for its function. a** Schematic representation of RCD1 domain structure containing a bipartite NLS, WWE, PARP-like and RST domains. Intrinsically disordered regions are marked as IDR1-4. **b** Early flowering phenotype of *rcd1* is not reverted by expression of RCD1ΔWWE-HA (ΔW), RCD1ΔPARP-like-HA (ΔP), RCD1ΔRST-HA (ΔR), and RCD1^S/T^IDR2^A^-HA in *rcd1* background. Picture shows 5-week-old plants of representative lines under standard growth conditions. **c** Wild-type MV sensitivity is not restored in lines expressing RCD1ΔWWE-HA (ΔW), RCD1ΔPARP-like-HA (ΔP), and RCD1ΔRST-HA (ΔR) constructs. PSII inhibition (Fv/Fm) by MV was measured in indicated lines using 0.5 μM MV. For each experiment, leaf discs from three individual rosettes were used. The experiment was performed three times with similar results. Mean ± SD are shown. * – *P* value < 0.05 with Tukey corrected *post hoc* test at the selected time point between *rcd1*: RCD1ΔPARP-like-HA and *rcd1*: RCD1-HA lines; *** – *P* value < 0.001 with Tukey corrected *post hoc* test at the selected time point between *rcd1*: RCD1ΔWWE-HA and *rcd1*: RCD1-HA lines. Source data and full statistics are presented in Supplementary Data 1. **d** Leaf shape phenotype of *rcd1* can be complemented by re-introduction of wild-type RCD1-HA, but not by RCD1 with mutated NLS (RCD1*nls*-HA) into the mutant background. The photo shows 3-week-old plant rosettes of two independent lines (A and B) for each construct under standard growth conditions. **e** RCD1 requires its NLS to complement the *rcd1*-specific MV tolerance. PSII inhibition (Fv/Fm) by methyl viologen (MV) was measured in indicated lines using 1 μM MV. For each experiment, leaf discs from three individual rosettes were used. The experiment was performed three times with similar results. Mean ± SD are shown. *** – *P* value < 0.001 with Tukey corrected *post hoc* test; n.s. – nonsignificant difference. Source data and statistics are presented in Supplementary Data 1.

RCD1 to these NBs (Fig. 2a). Disappearance of RCD1 from NBs was not due to low protein abundance since immunoblot analysis of the corresponding lines showed increased abundance of RCD1 in all deletion construct lines compared to wild type RCD1-Venus (Supplementary figure 3a).

The WWE domain has been predicted to serve as interaction domain between proteins and PAR. In different model systems it has been shown that PAR recruits PAR-binding proteins to NBs[37]. Therefore, we tested whether chemical inhibition of PAR synthesis with 3-methoxybenzamide (3MB) would influence the NB localization of RCD1-Venus. Indeed, in the 3MB-treated plants RCD1-Venus localized almost exclusively to the nucleoplasm (Fig. 2b), suggesting that PARP activity, and the subsequent presence of PARylated proteins in the nucleus was necessary for RCD1 to localize to NBs.

To examine whether RCD1 could be the plant equivalent of mammalian PAR readers, we used recombinant RCD1 to test its interaction with PAR in vitro (Supplementary Fig. 3b). Full-length RCD1-His protein was used in the in-vitro assays, as it was produced with higher purity than the GST-RCD1; all other proteins were expressed and purified as GST fusions to allow comparison to the control (Human WWE domain-GST fusion). As shown in Fig. 2c, the WWE domain of RCD1 alone, as well as full-length RCD1, interacted with purified PAR in dot blot assay. Consistently, the deletion variant of RCD1 lacking the WWE domain (GST-RCD1ΔWWE) did not bind PAR. For quantitative characterization of the RCD1-PAR interaction, we applied surface plasmon resonance (SPR) assays, which demonstrated a high-affinity interaction between PAR and RCD1 with a dissociation constant of 28.2 nM (Fig. 2d, Supplementary Fig. 3c). These binding properties are comparable to those described for animal WWE domains[10,38]. Consistent with the dot blot assay, the WWE domain was required for the association of RCD1 with PAR in SPR assays (Fig. 2d). Importantly, the interaction with PAR was

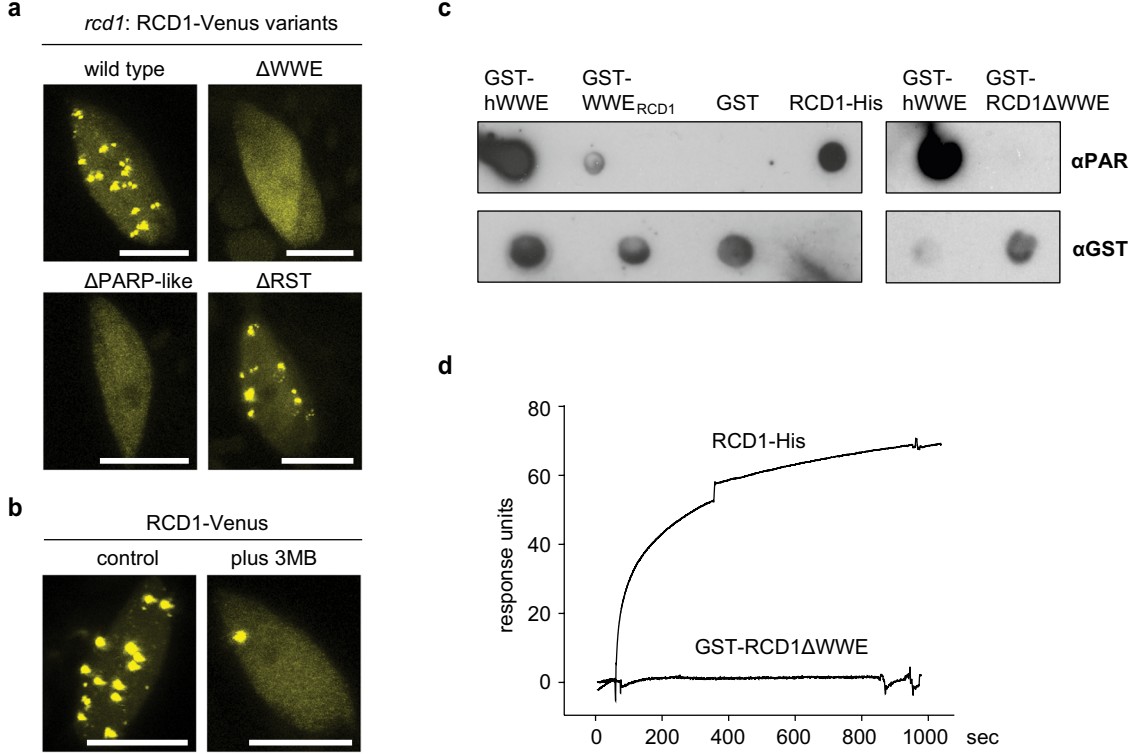

**Fig. 2 Localization of RCD1 to NBs and binding of PAR depends on WWE and PARP-like domains. a** Deletion of the WWE or PARP-like domains, but not the RST domain, prevents NB localization of RCD1. Confocal images were taken from stable Arabidopsis lines expressing full-length RCD1-Venus, RCD1ΔWWE-Venus, RCD1ΔPARP-Venus or RCD1ΔRST-Venus in the *rcd1* background. Scale bars indicate 10 μm. **b** NB localization of RCD1 is diminished by PARP inhibitor 3MB. Plants expressing RCD1-Venus were pretreated overnight at 4 °C without (control) or with 3MB, after which confocal microscopy was performed. Scale bars indicate 10 μm. **c** RCD1 binds PAR in vitro. PAR binding activity of immobilized GST-tagged domains of RCD1 and full-length RCD1-His was assessed by dot-blot assay using PAR-specific antibody. GST tagged human WWE domain (hWWE) and GST were used as positive and negative controls, respectively. GST antibody was used to assess protein loading. **d** WWE domain of RCD1 is required for interaction with PAR. SPR sensorgrams of interaction between immobilized RCD1-His or GST-RCD1ΔWWE and PAR profiled at 625 nM. Increase in response units shows association of PAR with RCD1-His but not with GST-RCD1ΔWWE.

specific since RCD1 did not interact with other structurally related compounds, such as monomeric ADP-ribose, or cyclic ADP-ribose (Supplementary Fig. 3d and e). Thus, our experiments showed that RCD1 bound PAR with high affinity and specificity, and the interaction required the WWE domain.

**RCD1 is phosphorylated by PPKs at multiple sites in the IDR2**. Hub proteins are often targets for multiple regulatory modifications allowing adjustment of their function in response to a variety of upstream signals. Since RCD1 has previously been reported to be an in vivo phosphoprotein[12], we analyzed the phosphorylation of immunoprecipitated RCD1 from protein extracts by mass spectrometry. We discovered 11 phosphopeptides harboring 30 potential phosphosites (Supplementary Table 1). All phosphopeptides identified in this and earlier studies are listed in Supplementary Table 1, and a schematic representation is shown in Supplementary Fig. 4. Notably, the phosphosites were enriched in the IDRs of RCD1 (Supplementary Fig. 4).

First we addressed the question, which kinases target RCD1. Our in vivo proteomic analyses of RCD1 interactors[5,12] revealed a recently described family of protein kinases, the PPKs. Here, we confirmed this interaction by targeted co-immunoprecipitation experiments, were PPK1, 3 and 4 were co-immunoprecipitated with RCD1 (Supplementary Fig. 5a). Furthermore, transient co-expression of RCD1-Venus and PPK-RFP in tobacco or in Arabidopsis seedlings demonstrated their co-localization in the afore-seen NBs (Fig. 3a and Supplementary Fig. 5b). In contrast,

expression of the Arabidopsis PPK-RFP constructs alone in tobacco showed uniform distribution of the proteins inside the nucleus (Fig. 3b). This suggests that a specific interaction with Arabidopsis RCD1 was required to recruit the Arabidopsis PPKs to NBs in transient expression in tobacco, since the tobacco SROs were not able to recruit the transiently expressed Arabidopsis PPKs to NBs.

To test for direct phosphorylation of RCD1 by PPKs, we used recombinant GST-tagged proteins for in vitro radiolabeling (Fig. 3c and d, Supplementary Fig. 6). The results showed that GST-PPK2 and GST-PPK4 phosphorylated GST-RCD1 directly in vitro. Mass spectrometric analysis of the in vitro phosphorylated GST-RCD1 revealed several PPK-dependent RCD1 phosphopeptides that were also identified in the in vivo pull-down experiments (Supplementary Table 1). Most of these phosphosites were clustered in the IDR2, the intrinsically disordered region between the WWE and PARP-like domains. We mutated the 15 identified phosphosites in this region to non-phosphorylatable alanines, yielding an RCD1 variant that is further referred to as RCD1$^{S/T}$IDR2$^A$. Phosphorylation of the GST-RCD1$^{S/T}$IDR2$^A$ by PPKs was abolished in in vitro (Fig. 3c, d).

**Phosphorylation of RCD1 by PPKs affects its function and localization to NBs**. To address the physiological role for phosphorylation of IDR2 by PPKs, we generated stable transgenic Arabidopsis lines expressing the RCD1$^{S/T}$IDR2$^A$-HA in *rcd1* background. Despite the high protein abundance (Supplementary Fig. 7), RCD1$^{S/T}$IDR2$^A$-HA did not fully complement the MV

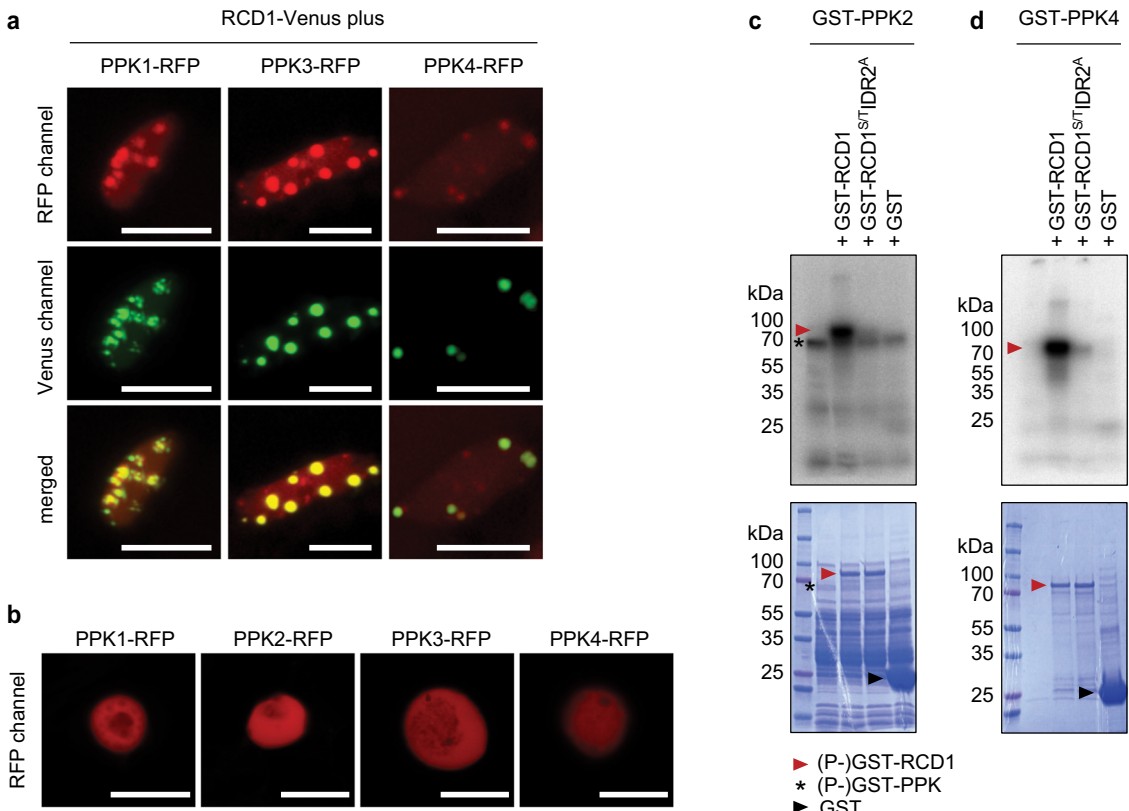

**Fig. 3 PPKs co-localize with RCD1 in NBs and directly phosphorylate RCD1 in vitro. a** RCD1 co-localizes with PPK1, PPK3 and PPK4 in NBs in Arabidopsis. RFP-tagged PPKs were transiently expressed in Arabidopsis seedlings expressing RCD1-Venus and the subnuclear localization was analyzed by confocal microscopy. Scale bars indicate 10 μm. **b** PPK-RFPs alone do not form NBs when transiently expressed in tobacco. PPK-RFP fusion proteins were expressed in tobacco leaves without co-expression of RCD1-Venus. **c, d** PPK2 and PPK4 directly phosphorylate RCD1. Recombinant GST-PPK2 and GST-PPK4 were used together with recombinant GST-RCD1 and the mutated GST-RCD1$^{S/T}$IDR2$^A$ protein in in vitro kinase assays. GST-PPK2 and 4 (asterisks) showed activity towards GST-RCD1 (red arrowhead). Upper panel shows autoradiograph, lower panel shows the Coomassie-stained SDS-PAGE.

tolerance of *rcd1* (Fig. 4a). The transgenic lines also still displayed *rcd1*-like abundance of AOX1/2 (Supplementary Fig. 7), the *rcd1*-type habitus (Fig. 1b) and flowering time (Supplementary Fig. 1c). This suggests that phosphorylation of IDR2 is involved in the regulation of RCD1 function. To address whether the phosphorylation of IDR2 affected RCD1 localization, we expressed RCD1$^{S/T}$IDR2$^A$-Venus in *rcd1* background. Similar to the wild-type RCD1, the mutated protein was localized to NBs, but intriguingly, the size and shape of the NBs were clearly different between the wild-type RCD1-Venus and the RCD1$^{S/T}$IDR2$^A$-Venus (Fig. 4b). Quantitative analysis of the images revealed that RCD1-Venus NBs were smaller but more abundant, whereas RCD1$^{S/T}$IDR2$^A$-Venus NBs were bigger but less abundant (Fig. 4b). In accordance with these data, expression of RCD1-Venus in triple *ppk* mutant background resulted in larger NBs compared to RCD1-Venus NBs in Col-0 (Supplementary Fig. 8). Altogether these data suggest that phosphorylation of the IDR2 of RCD1 is involved in the regulation of the RCD1 function by affecting its subnuclear distribution to NBs of different size and number, and presumably also function.

**Phosphorylation of RCD1 by PPKs regulates RCD1 stability.** Previous studies have shown that PPK-mediated phosphorylation affects stability of proteins by targeting them for degradation[32,33,35]. Accordingly, the non-phosphorylatable RCD1$^{S/T}$IDR2$^A$-HA had increased abundance compared to wild-type RCD1-HA in independent transgenic lines (Supplementary Fig. 7). The fact that RCD1$^{S/T}$IDR2$^A$ accumulated at increased abundance

(Supplementary figure 7) but did not complement *rcd1* phenotypes (Fig. 1b, Supplementary Fig. 1c, Fig. 4c and Supplementary Fig. 7) suggests that degradation of RCD1 is required for its physiological function. The triple *ppk* mutant plants also had increased native RCD1 protein levels (Fig. 4C), further suggesting that phosphorylation of RCD1 by PPKs affects its stability. Consistently, the higher accumulation of RCD1 in the triple *ppk* mutant coincided with lower resistance of plants to MV as compared to wild type (Fig. 4d). In order to form a concept regarding the regulation of RCD1 protein stability, we performed a cell-free degradation assay using recombinant GST-RCD1 and native extracts from wild-type Col-0 plants. Protease inhibitors targeting serine-, cysteine- and metalloproteases were added to the protein extracts in the reaction mix, yet GST-RCD1 was rapidly degraded. Addition of the proteasome inhibitor MG132 (Supplementary Fig. 9), however, could successfully prevent RCD1 degradation, which suggests that RCD1 protein level was specifically regulated by proteasomal degradation. Concluding, the results suggest that PPK-dependent phosphorylation of RCD1 plays an important regulatory role in RCD1 protein stability and function.

**Discussion**
The experiments described here have been designed to elucidate the mechanistic details of RCD1 function beyond the earlier studies addressing the role of RCD1 in transcriptional regulation through interaction with transcription factors. Our results here suggest that nuclear targeting and the WWE and PARP-like domains linked by the IDR2 play a fundamental role in the

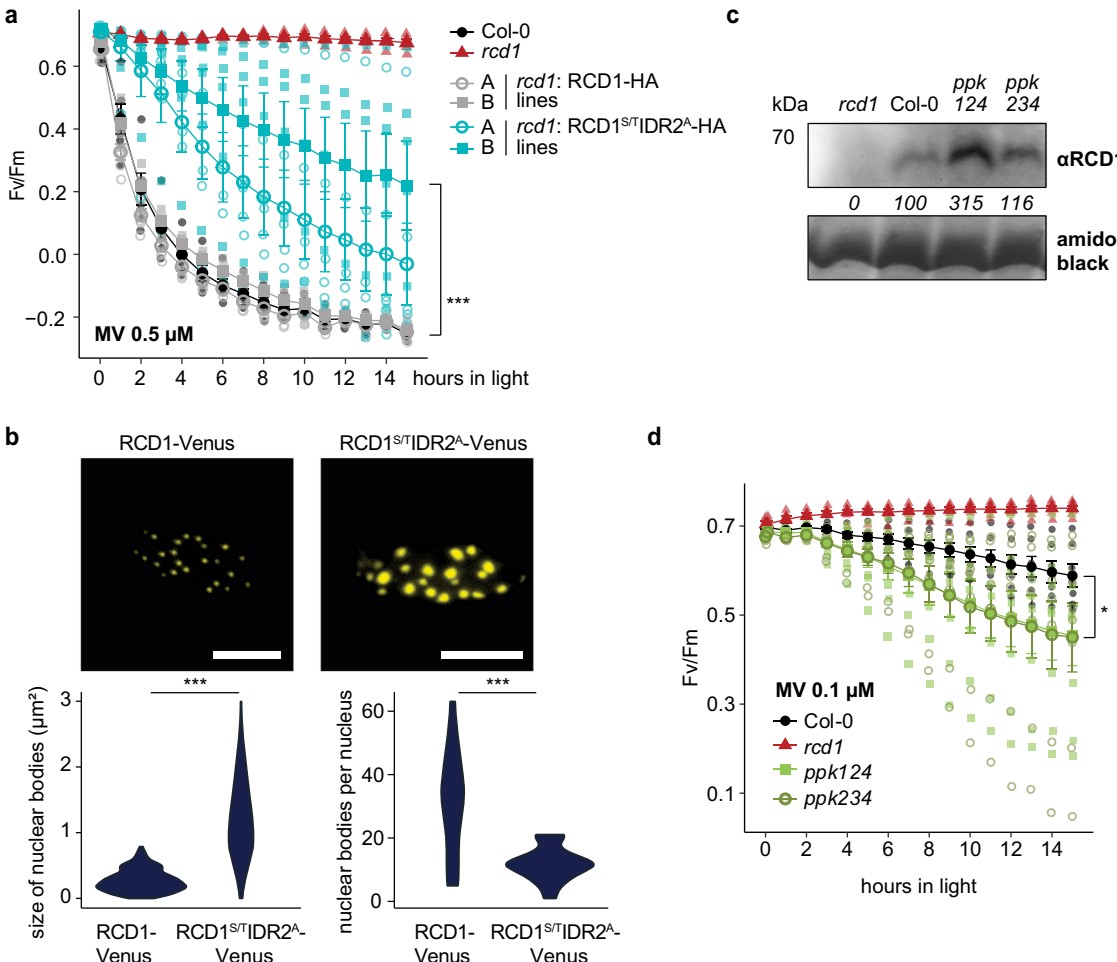

**Fig. 4 Phosphorylation by PPKs regulates RCD1 function and protein level. a** RCD1$^{S/T}$IDR2$^A$-HA variant does not fully complement *rcd1*-specific tolerance to MV. PSII inhibition (Fv/Fm) by MV was measured in indicated lines using 0.5 μM MV. For each experiment, leaf discs from at least four individual 3-week-old rosettes were used. The experiment was performed three times with similar results. Mean ± SD are shown. *** – P value < 0.001 with Tukey corrected *post hoc* test at the selected time point between *rcd1*: RCD1$^{S/T}$IDR2$^A$-HA (line A) and *rcd1*: RCD1-HA (line A) lines. Full source data and statistics are presented in Supplementary Data 1. **b** RCD1-Venus and RCD1$^{S/T}$IDR2$^A$-Venus are localized to distinct NBs. 10-day-old plate grown seedlings were soaked in 0.05% Tween-20 overnight at 4 °C and the subnuclear localization of RCD1-Venus and RCD1$^{S/T}$IDR2$^A$-Venus was analyzed by confocal microscopy. Representative images are shown. For quantification the images were acquired from 4 seedlings in each of which 4-5 nuclei from mesophyll cells were imaged as a Z-stack and then condensed by maximum projection. Violin plot is shown. (*** – P value < 0.001 one-way ANOVA). Scale bars represent 10 μm. Full source data and statistics are presented in Supplementary Data 1. **c** RCD1 accumulation in *ppk* triple mutants is higher than in Col-0. RCD1 level was assessed in total protein extracts from 3-week-old plants by immunoblot analysis with RCD1-specific antibody. The signal was quantified using ImageJ. The abundance in percent relative to Col-0 (100%) is shown under the immunoblot panel. Rubisco large subunit detected by amido black staining is shown as a control for equal protein loading. **d** *ppk* triple mutants are more sensitive to MV than Col-0. PSII inhibition (Fv/Fm) by MV was measured in indicated lines using 0.1 μM MV. For each experiment, leaf discs from four individual 3-week-old rosettes were used. The experiment was performed three times with similar results. Mean ± SD are shown. * – P value < 0.05 with Tukey corrected post hoc test at the selected time point between *ppk124* and Col-0. Source data and statistics are presented in Supplementary Data 1.

physiological function of RCD1. Our data also suggests that the WWE-IDR2-PARP-like module acts as a PAR-dependent protein-interaction platform, which is regulated by phosphorylation. The WWE- and PAR-dependent localization of RCD1 to NBs suggests that one function of this module is to determine the subnuclear location of RCD1. It has been suggested that in animal cells the PAR polymer provides an interaction platform for PAR-readers to modulate cellular responses, including chromatin remodeling, protein degradation and cell death[16] and several of these PAR-related processes co-localize with PAR-binding proteins in NBs[38]. In plants, however, the small number of confirmed PARylated proteins identified so far does not contain any proteins that possibly could recruit RCD1 to NBs. The recently published models on the roles of PAR and phosphorylation of intrinsically

disordered proteins in subnuclear compartmentalization via liquid-liquid phase separation (LLPS)[39,40] provides also new hypotheses for further studies to elucidate the activity and potential role of RCD1 in the regulation NB formation.

RCD1 and SRO1 are the only Arabidopsis proteins possessing a WWE domain and their domain architecture is reminiscent of human PARP11 and PARP 14, which also contain both WWE- and PARP domains. While human PARP11 and PARP14 have been shown to act as MARTs, the Arabidopsis RCD1 and SRO1 proteins have not shown either PARP or MART activity[8,12,13]. The wheat Ta-SRO1 protein has initially been reported to have PARP activity[41] but recent data[42] argue against the canonical PARP activity of Ta-SRO1. However, the SRO protein family members do also have a functional connection with PARylation

since the Arabidopsis SRO2 protein has recently been suggested to act as a MART[13]. The evolutionarily early and conserved acquisition of the plant-specific RST domain in SROs of all land plants[8], as well as its binding to plant-specific transcription factors suggest the evolution of a unique functional element in plant transcriptional regulation. The similar, evolutionarily early and conserved acquisition of the WWE-IDR2-PARP-like domain structure of the type A SROs (RCD1 and SRO1 in Arabidopsis) in all land plants[8] suggests an indispensable function for these proteins. Accordingly, the Arabidopsis *rcd1 sro1* double mutant exhibits severe developmental defects in embryogenesis, or is in most cases not viable at all[4,43], showing that RCD1/SRO1, which form both homo- and heterodimers[12], are essential proteins in Arabidopsis – a functional form of at least one of them must be present in cells. Consequently, our data suggests that the evolution of RCD1-type proteins represents a unique, indispensable plant-specific de-coding system for PARylation (PAR readers) involved in transcriptional regulation.

Phosphorylation events in the IDR2 play an important role in the regulation of RCD1. We have shown earlier that IDR2 is involved in the homo- or heterodimerization of RCD1 and SRO1, and is targeted by the oomycete effector protein HaRxL106[12] further highlighting the role for PARylation in plant immunity[13,24,27,28]. Here our results suggest that multi-site phosphorylation of the IDR2 by PPKs affects RCD1 function, NB localization, and stability. Protein kinases preferentially target IDRs due to their high accessibility[44]. Phosphorylation at single or multiple sites, either sequentially or combinatorically, changes the properties of the protein in different ways. This makes IDRs highly sensitive switches that are triggered by the level of phosphorylation[45]. Phosphorylation can influence the assembly of intrinsically disordered proteins into subnuclear structures through the formation or dissolution of membrane-less compartments, related to LLPS[39,40]. Distinct NBs in case of the wild type RCD1 and RCD1$^{S/T}$IDR2$^A$ support such a role of IDR2 phosphorylation, suggesting different functionality of these NB.

Multiple phosphorylation can trigger disorder-to-order transitions of the protein structure, affect protein-protein interactions, and protein degradation[46]. For example, it has been shown that progressive multiple phosphorylation of IDRs in yeast cell cycle-regulating proteins controls their degradation[47]. It can be envisaged that the phosphorylation of the IDR2 of RCD1 by PPKs in multiple sites could represent a similar regulatory function. Furthermore, experiments reported here suggest that phosphorylation of IDR2 is critical for the function of RCD1 as a negative regulator of transcription factors it interacts with, including DREBs and ANAC013/017[5]. It is known that DREB2A is tightly regulated via proteasomal degradation[48], its interaction with RCD1 might represent another layer of posttranslational control. Negative regulation of these transcription factors by RCD1 would keep the expression of stress-related genes under control in, for instance, unstressed conditions.

The other kinases that target IDR2 likely modulate other functions. For example, we found that several GSK3/Shaggy-like protein kinases (ASKs) target RCD1 specifically at Thr 204, which is located in IDR2 (Supplementary Fig. 10a and b), without affecting the stability of RCD1. It will be of interest for future studies to investigate the different outcomes of regulation by single-site phosphorylation in IDR2. Overall, the presence of four IDRs in RCD1 and the multiple phosphorylation sites within them allow a large spectrum of modulation and fine-tuning of downstream responses.

Analysis of WWE domain proteins in animals has shown that the WWE domains co-exist in proteins not only with PARP/PARP-like domains, but also with E3 ubiquitin-ligase domains[9,10], which are involved in proteasomal degradation processes. For example, the human E3 ligase RNF146 interacts with PARylated substrates via its WWE domain, and mediates their ubiquitination for proteasomal

degradation[10,11]. In plants, however, the WWE domain occurs exclusively in RCD1 and SRO1 and does not appear to be directly linked to E3 ubiquitin ligases. Intriguingly, however, a large fraction of transcription factors interacting with RCD1 are known to be regulated by proteasomal degradation[33,48,49]. Furthermore, RCD1 itself seems to be degraded by the proteasome (Supplementary Fig. 9), several proteins related to ubiquitin-dependent protein catabolism co-immunoprecipitated with RCD1[5] and gene ontology analysis of altered gene expression in the *rcd1* mutant revealed enrichment in ubiquitin-proteasome-pathway associated genes[4]. Recently two publications[13,28] showed the importance of PARylation and MARylation in plants in the context of protein ubiquitination and degradation, highlighting their involvement in plant immunity. These data support that the functional link between PARylation, ubiquitination, and proteasomal degradation is evolutionary conserved also in plants and apparently RCD1 as likely PAR-reader also plays a central role within this signaling network. Notably, the WWE-PARP-like-RST domain architecture is strictly conserved in type-A SRO proteins in monocots and dicots[8]. Therefore, our findings on the regulatory roles of the RCD1 WWE-IDR2-PARP-like module is going to have important implications for translational research, for example understanding the role of rice and wheat SRO proteins in drought and salinity stress tolerance[41,50].

Concluding, we suggest RCD1 is the plant PAR-reader and assume the following scenario as RCD1's mode of action (Fig. 5): RCD1 is targeted to the nucleus (#1) where it is recruited to NBs by binding to PARylated proteins via its WWE-IDR2-PARP-like

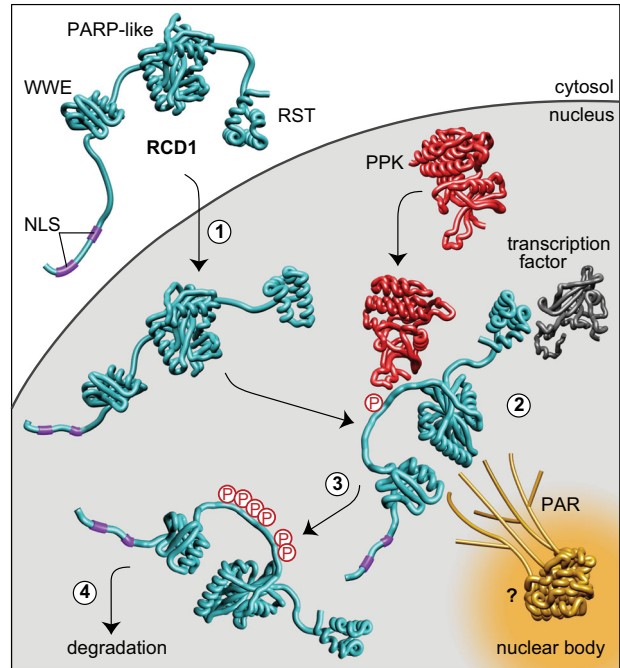

**Fig. 5 A model for the regulation of nuclear RCD1 function by PAR binding and phosphorylation by PPKs.** (1) RCD1 enters the nucleus by means of its bipartite N-terminal NLS sequence. In the nucleus, RCD1 interacts with PPKs, with diverse transcription factors and with PAR (2). PAR recruits RCD1 to NBs of yet uncharacterized nature. Unknown PARylated proteins involved in RCD1 recruitment are labeled with a question mark. RCD1 is phosphorylated by PPKs at multiple sites in IDR2 (3), which affects RCD1 functions and facilitates RCD1 degradation (4). RCD1 structure was predicted in RaptorX (http://raptorx.uchicago.edu/). Structural model of the WWE domain is based on mouse RNF146 (2RSF), structures of RCD1 PARP-like (5NGO)[12] and RST (5N9Q)[1] domains have been reported. Terminal and inter-domain regions of RCD1 are not drawn to scale.

module. In addition, RCD1 binds diverse TFs with its RST domain (#2). Phosphorylation of IDR2 by PPKs (#3) affects the NB localization and facilitates degradation of RCD1 and possibly also its partner TFs (#4). Thus, it is necessary for the function of RCD1 as a negative transcriptional co-regulator. This model proposes a new mechanism of fine-tuning transcriptional regulation, involving PAR-dependent subnuclear compartmentalization and post-translational modification of the hub-protein RCD1.

## Methods

**Plants, mutants and chemical treatments.** *Arabidopsis thaliana* plants were grown on soil (peat: vermiculite = 1:1) under white luminescent light (220-250 $\mu$mol m$^{-2}$ s$^{-1}$) at a 12-hour photoperiod and 22/18 °C. Seedlings were grown for 10 days on 1 x MS basal medium (Sigma) with 0.5 % Phytagel (Sigma). Arabidopsis *rcd1-4* mutant (Gabi-Kat line GK-229D11) was used as a background for all complementation lines. The *ppk* triple mutants were kindly provided by Dr Dmitri Nusinow (Donald Danforth Plant Science Center, St. Louis)[31]. Treatments with chemicals methyl viologen (MV, 0.1, 0.5, or 1 $\mu$M, as indicated in the figures) and 3-methoxybenzamide (3MB, Sigma, 10 mM) were performed on leaf discs floating on Milli-Q water supplemented with 0.05% Tween 20 (Sigma), overnight at room temperature or at 4 °C, accordingly. For 4′,6-diamidino-2-phenylindole (DAPI) staining, the seedlings were vacuum-infiltrated with 0.1 mM DAPI in Milli-Q water supplemented with 0.05% Tween 20.

**Plasmids.** Full-length AtRCD1, the WWE-domain (amino acids 1-155), RCD1$\Delta$WWE (consisting of PARP- and RST-domains, amino acids 241-589), RCD1$\Delta$PARP (missing the residues 304-443) and the C-terminal part of RCD1, including the RST-domain (amino acids 468-589), were cloned into pGEX4T-1 for N-terminal GST fusion using primers listed in Supplementary Table 2. Full-length AtRCD1 was also cloned into the pET8c vector for N-terminal His-fusion[8]. For generating N-terminal GST-fusion constructs, PPK1-4 cDNAs were cloned into pGEX6P-1, and ASK cDNAs into pGEX4T-1. The kinase-dead ASK loss-of-function constructs contain a Lys-Arg mutation in the kinase activation loop.

For generating fusion constructs of RCD1 (GST, HA & Venus) where the IDR2 is nonphosphorylateable (GST-RCD1$^{S/T}$IDR2$^A$), all phospho-serine and phospho-threonine residues within IDR2 were mutated to alanine residues by gene synthesis (Genescript Biotech, Netherlands). This IDR2 was cloned into the respective wild-type full-length RCD1 construct using end-joining In-Fusion (Takara Bio) using primers listed in Supplementary Table 2.

To generate the RCD1-Venus construct, RCD1 cDNA was fused to the *UBIQUITIN10* promoter region and to the C-terminal triple Venus YFP tag in a MultiSite Gateway reaction[51]. The $\Delta$WWE (missing the residues 90-151), $\Delta$PARP (missing the residues 304-443) and $\Delta$RST (missing the residues 462-589) deletions were introduced by PCR using primers listed in Supplementary Table 2 and end-joining using In-Fusion.

Construction of transgenic lines expressing HA-tagged RCD1 (RCD1-3xHA) is reported[4]. RCD1*nls*-HA variant was generated by substituting the basic amino acids of the NLS with aliphatic ones (K21L/R22I and R56I/R57I), using the vector pDONR/Zeo that contained the RCD1 promoter followed by the wild-type genomic RCD1 sequence[4]. PCR was performed with Q5 High-Fidelity DNA Polymerase (New England Biolabs) and the primers listed in Supplementary Table 2. After sequential mutation of the two parts of the bipartite NLS, the construct was transferred to the Gateway pGWB13 binary vector and introduced into the plants by the floral dip method. The $\Delta$WWE (missing the residues 90-151), $\Delta$PARP (missing the residues 304-443) and $\Delta$RST (missing the residues 462-589) deletions were introduced by PCR using primers listed in Supplementary Table 2 and end-joining using In-Fusion.

For generating epitope-tagged PPK fusions, the coding sequences of the four *PPK* genes lacking their stop codons were cloned into NcoI/XhoI-digested pENTR4 using In-Fusion enzyme (Takara). The *PPK* coding sequences were then recombined by Gateway® Clonase II (Invitrogen) reactions into pB7WGR2[52] or pGWB414[53] to create RFP and 3xHA-tagged variants, respectively.

**Spectroscopic measurements of photosynthesis.** Chlorophyll fluorescence was measured by MAXI Imaging PAM (Walz, Germany) essentially as described[5]. PSII photoinhibition protocol consisted of repetitive 1-hour periods of blue actinic light (450 nm, 80 $\mu$mol m$^{-2}$ s$^{-1}$) each followed by a 20-min dark adaptation, then $F_o$ and $F_m$ measurement. PSII photochemical yield was calculated as $F_v/F_m = (F_m-F_o)/F_m$. The assays were performed in 96-well plates. In each assay leaf discs from at least 4 individual plants were analyzed. Each assay was reproduced at least three times.

**SDS-PAGE and immunoblotting.** For immunoblotting of total plant extracts, the plant material was frozen immediately after treatments in liquid nitrogen and ground. Total proteins were extracted in SDS extraction buffer (50 mM Tris, pH 7.8, 2 % SDS, 1 x protease inhibitor cocktail; P9599, Sigma), 2 mg/ mL NaF) for 20 min at 37 °C and centrifuged at 18,000x *g* for 10 min. Supernatants were normalized for protein concentration and resolved by SDS-PAGE. After electrophoresis, proteins were

electroblotted to PVDF membrane and probed with specific antibodies: $\alpha$HA (Roche), $\alpha$GFP (Milteny Biotech), $\alpha$GST (Sigma), $\alpha$PAR (Trevigen), $\alpha$RCD1[5], and $\alpha$AOX1/2 (Agrisera AS04 054). The signal was visualized by ECL Prime chemiluminescence reagents (GE Healthcare). Quantification of the signal was done using ImageJ.

**Confocal microscopy.** The subcellular localization of RCD1 in stable expression Arabidopsis line was analyzed by confocal microscopy with a Leica SP5 II HCS inverted microscope using a solid-state blue laser for visualizing YFP and chloroplast autofluorescence (detection with 521–587 and 636–674 nm range, respectively). For co-localization studies of RCD1-Venus and PPK-RFP fusion constructs, the binary plasmids were transformed into *A. tumefaciens* strain GV3101 pMP90. Proteins were transiently expressed in *N. benthamiana* leaves as described below for co-immunoprecipitation assays. YFP was excited using a 488 nm laser with a detection window of 519–556 nm and RFP was excited using a 561 nm laser with detection at 599–657 nm.

**Protein expression and purification.** Fusion proteins were expressed in *E. coli* BL21 (DE3) Codon Plus strain. LB medium supplemented with ampicillin (100 $\mu$g ml$^{-1}$) and chloramphenicol (50 $\mu$g ml$^{-1}$) was inoculated with overnight bacterial culture and grown at 37 °C until OD$_{600}$ reached 0.6–0.8. Induction of protein expression was done by addition of isopropyl-$\beta$-D-galactoside (IPTG) to a final concentration of 0.2–0.5 mM, and the culture was transferred to 28 °C. Cells were harvested after 4 hours by centrifugation at 6000 g and stored at -20 °C.

The cell pellets were resuspended in a lysis buffer consisting of 50 mM Tris-HCl, pH 7.5, 150 mM NaCl, 5 mM DTT, protease inhibitors cocktail (Complete, Roche Diagnostics). Lysozyme was used to lyse the cells to a concentration of 0.2 mg ml$^{-1}$ and incubation for 20 min at 4 °C with gentle shaking. DNaseI (0.02 mg ml$^{-1}$) was used to digest DNA in presence of 10 mM MgCl$_2$. The cell lysates were clarified by centrifugation at 20,000 g for 15 min at 4 °C. The GST-tagged proteins were purified by affinity chromatography using GSH beads (GE Healthcare) according to manufacturer's instructions.

For RCD1-His purification inclusion bodies were resuspended in solubilization buffer (50 mM Tris-HCl, pH 8.0, 0.5 M NaCl, 6 M guanidine hydrochloride (GuHCl), 5 mM imidazole, 10 mM 2-mercaptoethanol) and solubilized by stirring overnight at 4 °C. After centrifugation for 20 min at 15.000 g the supernatant was loaded onto two 1-ml HisTrap columns (GE Healthcare) which were pre-equilibrated with solubilization buffer. ÄKTA Prime plus chromatography system (GE Healthcare) was used for on-column refolding. The column was first washed with 20 column volumes (CV) of solubilization buffer and subsequently with 20 CV of solubilizations buffer where 6 M GuHCl was substituted by 6 M urea. Refolding was done by running linear urea gradient from 6 M to 0 M with flow rate 0.5 ml min$^{-1}$ within 20 CV. Refolded protein was eluted with elution buffer (50 mM Tris-HCl, 0.5 M NaCl, 1 mM 2-mercaptoethanol and 0.5 M imidazole) and desalted to 50 mM Tris-HCl, pH 8.0, 150 mM NaCl, 10% glycerol using HiTrap desalting column (GE Healthcare).

**Poly(ADP-ribose) dot-blot assay.** Purified His and GST fusion proteins or GST alone (500 ng) were blotted onto nitrocellulose membrane (BioRad). The nitrocellulose membrane was rinsed with TBS-T buffer (10 mM Tris-HCl at pH 7.4, 150 mM NaCl and 0.05 % Tween 20) three times. The membrane was incubated with 100 nM of purified PAR (Trevigen, 4336-100-01, 10 $\mu$M stock, polymer size 2-300 units) for 1 h at room temperature. After 5 washes with TBS-T and TBS-T containing 1 M NaCl, the membrane was blocked with 5 % milk followed by immunoblotting with mouse $\alpha$PAR (Trevigen) or $\alpha$GST (Sigma) antibody.

**Surface plasmon resonance.** Recombinant RCD1-His and GST-RCD1$\Delta$WWE proteins were coupled to a Biacore CM5 sensor chip (GE Healthcare) *via* amino-groups. PAR (625 nM) (Trevigen) was profiled at a flow rate of 30 mL/min for 300 s, followed by 600 s flow of wash buffer (10 mM HEPES, pH 7.4, 150 mM NaCl, 3 mM EDTA, 0.05% Surfactant P20). Mono ADP-ribose (Sigma) and cyclic ADP-ribose (Sigma) were profiled at 1 mM concentration. After analysis in BiaEvalution (Biacore, GE Healthcare), the normalized resonance units were plotted over time with the assumption of one-to-one binding.

**Transient protein expression in *N. benthamiana* and Arabidopsis.** Binary vectors harbouring RCD1-GFP or PPK-3xHA fusions were transformed into *A. tumefaciens* strain GV3101 pMP90. For expression, Agrobacteria were scraped from selective YEB plates and resuspended in infiltration medium (10 mM MES pH 5.6, 10 mM MgCl$_2$) and the OD$_{600}$ was adjusted to 0.8. To suppress transgene silencing, Agrobacteria expressing the tomato bushy stunt virus 19 K silencing suppressor were co-infiltrated. After adding acetosyringone (Sigma) to a final concentration of 100 $\mu$M and incubation for 2 h at room temperature, Agrobacteria were mixed in a ratio of 1:1:2 (19 K) and infiltrated in *N. benthamiana* leaves.

For transient Arabidopsis expression the FAST co-cultivation technique was used[54]. In short binary vectors harbouring PPK-RFP fusions were transformed into *A. tumefaciens* strain GV3101 pMP90. From overnight liquid LB-culture Agrobacteria were washed and resuspended in co-cultivation medium to OD$_{600}$ 2.5. RCD1-Venus seedlings grown for 5 days in long days (16/8, light/dark) were soaked in Agrobacteria containing co-cultivation medium for 20 minutes.

**Co-immunoprecipitation.** Infiltrated leaf tissue was harvested 72 h later and proteins were extracted by grinding leaf tissue in liquid nitrogen followed by resuspension in extraction buffer (50 mM Tris-HCl pH 7.5, 150 mM NaCl, 10% glycerol, 1 mM EDTA, 5 mM DTT, 1x protease inhibitor cocktail [P9599, Sigma], 10 μM MG132) at a ratio of 2 mL / g FW. Protein extracts were centrifuged at 20000 x g / 4 °C/20 min and a fraction of the supernatant was saved as input sample. 15 μL of αGFP-nanobody:Halo:His6 magnetic beads[55] were added to 1.5 mL of protein extract followed by incubation on a rotating wheel at 4°C for 5 min. The beads were washed 3 times with 1 mL extraction buffer using a magnetic tube rack and then boiled in 80 μL SDS sample buffer to elute protein from the beads. For immunoblots, protein samples were separated by SDS-PAGE and electro-blotted onto PVDF membrane. Antibodies used were αGFP (210-PS-1GP, Amsbio) and αHA (Roche).

**Kinase activity assays.** In vitro kinase assays using recombinant proteins were performed in a total volume of 20 μL of kinase buffer (20 mM HEPES, pH 7.5, 15 mM MgCl$_2$, and 5 mM EGTA). The reaction was started with 2 μCi [γ-$^{32}$P] ATP and incubated at room temperature for 30 min. The reaction was stopped by the addition of 5 μL of 4x SDS loading buffer. Proteins were resolved by SDS-PAGE and the gel was dried and exposed overnight to a phosphor imager screen. For the kinase activity test, GST-PPKs were tested against 5 μg myelin basic protein (MBP; Sigma) and 5 μg Casein in 0.1 M Tris pH 8.8 (Sigma). For identification of in vitro phosphorylation sites by LC-MS/MS, 1.5 mM unlabeled ATP was used in the kinase buffer. The proteins were separated by SDS-PAGE, followed by Coomassie Brilliant Blue staining and were digested by trypsin (Promega). Only GST-PPK2 and GST-PPK4 could be expressed and purified from *E. coli* with detectable kinase activity (Supplementary figure 6).

**Cell-free degradation assays.** For cell-free degradation assays proteins were extracted in 25 mM Tris-HCl, pH 7.5, 50 mM NaCl, 10 mM MgCl$_2$, 10 mM ATP, and 10 mM DTT. After centrifugation (14,000 g, 10 min, 4 °C) supernatants were incubated at room temperature with recombinant GST-RCD1 and the reactions terminated with SDS sample buffer and incubation at 70 °C (10 min). Inhibitors (50 μM MG132, 4 mM PMSF, 50 μM leupeptin, 50 μM aprotinin, and 50 μM pepstatin A) were applied using 0.2% DMSO as vehicle. The proteins were separated by SDS-PAGE and analyzed by Western blot as described above.

**LC-MS/MS.** Phosphopeptides were enriched from tryptic digests using TiO$_2$ microcolumns (GL Sciences Inc., Japan). Enriched phosphopeptides were analyzed by a Q Exactive mass spectrometer (Thermo Fisher Scientific) connected to Easy NanoLC 1000 (Thermo Fisher Scientific). Peptides were first loaded on a trapping column and subsequently separated inline on a 15-cm C18 column (75 μm × 15 cm, ReproSil-Pur 5 μm 200 Å C18-AQ, Dr. Maisch HPLC). The mobile phase consisted of water with 0.1% (v/v) formic acid (solvent A) or acetonitrile/water (80:20 [v/v]) with 0.1% (v/v) formic acid (solvent B). A linear 60-min gradient from 6 to 42% (v/v) B was used to elute peptides. Mass spectrometry data were acquired automatically by using Xcalibur 3.1 software (Thermo Fisher Scientific). An information-dependent acquisition method consisted of an Orbitrap mass spectrometry survey scan of mass range 300 to 2000 m/z (mass-to-charge ratio) followed by higher-energy collisional dissociation (HCD) fragmentation for 10 most intense peptide ions. Raw data were searched for protein identification by Proteome Discoverer (version 2.4) connected to in-house Mascot (v. 2.6.1) server. Phosphorylation site locations were validated using phosphoRS algorithm. A SwissProt database (https://www.uniprot.org/) was used with a taxonomy filter Arabidopsis. Two missed cleavages were allowed. Peptide mass tolerance ± 10 ppm and fragment mass tolerance ± 0.02 Da were used. Carbamidomethyl (C) was set as a fixed modification and Met oxidation, acetylation of protein N-terminus, and phosphorylation of Ser and Thr were included as variable modifications. Only peptides with a false discovery rate of 0.01 were used.

**Statistics and Reproducibility.** The statistical analyses were performed using IBM SPSS Statistics software (version 28.0.1.0). Tukey corrected *post hoc* test was used for analysis of chlorophyll fluorescence data with significance reported at P-value < 0.001 or < 0.05. Microscopy images were quantified using ImageJ and statistical analysis was performed using R (version 4.0.5, R Foundation for Statistical Computing). Datapoints outside 1.5 interquartile range were deemed outliers and excluded, the remaining data were used to perform one-way ANOVA. All source data and statistical tests are reported in the Supplementary Data 1.

**Reporting summary.** Further information on research design is available in the Nature Portfolio Reporting Summary linked to this article.

## Data availability

The mass spectrometry-based proteomics data generated during this study are available from the Proteome Xchange Consortium via the PRIDE partner repository[56] with the dataset identifier PXD039877. Uncropped and unedited Western blots/gel images are shown in the Supplementary figure 11. The source data used for statistical analysis are reported in Supplementary Data 1. All other data are available from the corresponding author on reasonable request.

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

## Acknowledgements

We thank Damien Marchese, Shaimaa Reda and Dr. Melanie Carmody for the help in generating the transgenic lines. We thank Dr. Dmitri Nusinow for providing the seeds of *ppk* triple mutants. We thank Maria Aatonen and Maria Semenova for help in Biacore experiments which were performed at the Biomolecular Interaction Unit, Faculty of Biological and Environmental Sciences, University of Helsinki. We thank Mika Molin and Marko Crivaro for help with confocal microscopy at the Light Microscopy Unit of the Institute of Biotechnology, University of Helsinki. We thank Prof. Romy Schmidt-Schippers for her input in studying the RCD1 nuclear bodies. Mass spectrometry analyses were performed at the Turku Proteomics Facility, supported by Biocenter Finland. This work was supported by the University of Helsinki (JK); the Academy of Finland Centre of Excellence programs (2006-11; and 2014-19; JK) and Research Grant (Decision 250336; JK). MW acknowledges funding from the Academy of Finland (Decisions 275632, 283139, 312498, and 323917). LW acknowledges funding by the German Research Foundation (DFG; grant WI 3670/2-1) and core funding from the Leibniz Institute of Plant Biochemistry (IPB) and the FU Berlin Dahlem Centre of Plant Sciences. RG acknowledges funding from the Doctoral Programme in Plant Sciences by the University of Helsinki and Ella and Georg Ehrnrooth Foundation.

## Author contributions

J.V., R.G., J.K.W., C.J., L.W., M.W., A.S. and J.K. conceived and designed experiments. J.V., R.G., J.K.W., R.D.M., I.D., T.P., N.B., A.S. and L.W. carried out experiments. J.V., R.G., J.K.W., R.D.M., I.D., N.B., C.J., L.W., M.W., A.S. and J.K. analyzed the data. J.V., R.G., J.K.W., L.W., A.S. and J.K. wrote the article. All authors read and contributed to the final article.

## Competing interests

The authors declare no competing interests.
