## [Peer Review File · Communications Biology]

Reviewers' comments:

Reviewer #1 (Remarks to the Author):

The Manuscript by Vainonen et al. addresses the functional roles of the WWE and PARP-like domains in RCD1 (and SRO1), which have so far been mostly elusive compared to that of the TF-interacting RST domain. The work pushes the understanding of RCD1 in a new direction, which is both refreshing and important. A number of issues should be addressed:

- 1) The abstract does not really report much about the findings of the work, but rather gives an introduction,. This should be changed to invoke interest.
- 2) RCD1 is referred to as an intrinsically disordered protein. I would call it a multi domain protein with IDRs.
- 3) Fig. 2B: Some NB localization retained?
- 4) Fig. 2C: When are GST and when are His-tagged proteins used? Why no anti-GST detection for GST-hWWE at left?
- 5) Fig. 2D: Why are different recombinant proteins used? This makes control experiments needed (GST and His tags alone). Furthermore, protein is immobilized via amino-groups, which may affect both protein structure and interactions. This should be addressed.
- 6) Line 163: Here GST-RCD1 (and not His-RCD1) is used. Again what is used when and why?
- 7) Model presented in figure 4 is very interesting. However, it becomes very speculative when it comes to E3 ubiquitin-degradation. Could this be bioinformatically or experimentally addressed?
- 8) Can the authors relate previous own work on the negative regulatory effects of RCD1 on DREB2 and NAC TFs more directly to this work?

Reviewer #2 (Remarks to the Author):

PARYlation is a reversible post-translational modification that modulates diverse physiological responses in eukaryotes. PARYlation is mediated by poly(ADP-ribose) polymerase (PARPs). Plants have a group of unique PARP-like proteins, namely RCD1 and SROs. However, RCD1 and SROs do not have detectable PARP activities. This study shows that RCD1 functions as a PAR reader that directly binds PAR. RCD1 localizes to nuclear bodies in a PAR-dependent manner. The authors then identified PPKs as the kinases that phosphorylate RCD1 and regulate RCD1 abundance. This is the first report of a plant PAR reader, and elucidates its regulatory mechanism by protein phosphorylation. Together with a recent study about SRO2 possessing MARYlation activity that regulates protein stability, these studies expand the mechanistic understanding of plant-specific PARP-like protein functions. The manuscript was well written and easy to follow. The data are of high quality.

One point could be better explained. PPK-mediated phosphorylation reduces RCD1 abundance but is also required for RCD1 function. This seems to conflict.

In Fig. 3D, why GST-PPK was not labeled, and no phosphorylation was observed?

Please also discuss whether phosphorylation mutant RCD1S/TIDR2A affects NB localization.

Line 200, "The" should be "the".

Reviewer #3 (Remarks to the Author):

RCD1 is a disordered hub protein that has been previously shown to interact with over 30 TFs via its RST-domain. However, RCD1 also possesses other domains (WWE and PARP-like) that have been demonstrated to be functionally involved in facilitating PARylation and MARTylation within animal systems. Although no prior evidence has shown RCD1's involvement in these types of processes, the presence of these domains suggest that RCD1 may be involved in similar processes. Here, the authors excitingly propose that RCD1 possibly functions as a PAR reader by which RCD localizes to nuclear bodies by binding PAR via its WWE domain. Additionally, they show that PPKs co-localize in NBs with RCD1 to phosphorylate RCD1 and control its levels. Assuming the authors address the comments below, I recommend this paper for publication.

Major comments:

Does RCD1 actually interact with PARylated proteins? The authors use chemical analysis to show that PAR synthesis influences NB formation, However, in their Mass Spec analysis, do they find PARylated proteins that interact with RCD1 and if so, do these proteins localize to the same bodies?

Additionally, the test RCD1's ability to act as a PAR reader in vitro. However, this doesn't prove functionality as a PAR reader in vivo. Ideally, the authors should do a WWE domain swap and see if the plant and animal WWE domains are functionally interchangeable. I would suggest that if the authors cannot do this experiment, that they tone down their interpretation of RCD1 as a PAR reader.

The authors use a transient assay to show that PPKs require RCD1 for their localization to nuclear speckles. However, since the authors have the *rcd1* mutant and fluorescently-tagged PPK1 proteins, they should confirm that PPKs require RCD1 for localization by expressing PPKs in the *rcd1* mutant background. Additionally, as the authors have the triple *ppk* mutants, they should test if PPKs are necessary for RCD1 localization. Instead the only elude to this using a transient system (sup fig 5B). Disordered proteins are concentration dependent and since transient assays often OX protein, it is hard to say if this is a meaningful result.

Minor comments:

Figure 1D: The authors propose that the NLS signal is important for the leaf curling phenotype displayed in the *rcd* mutants but it is not sufficiently captured in this image. Please provide a better image, in which the phenotype is sufficiently observed or downplay this result.

Supplemental 3A: The authors mention that 100µg of protein was added per lane but they do not have the necessary endogenous controls showing this is true. Please add an endogenous control.

minor editing suggestions:

Line 85: "trough" needs to be replaced with "through"

Line 86: and an "a" between "in" and "non"

Please see below our detailed response to the points raised by reviewers (the reviewers' questions are marked italics).

Reviewers' comments:

Reviewer #1 (Remarks to the Author):

The Manuscript by Vainonen et al. addresses the functional roles of the WWE and PARP-like domains in RCD1 (and SRO1), which have so far been mostly elusive compared to that of the TF-interacting RST domain. The work pushes the understanding of RCD1 in a new direction, which is both refreshing and important. A number of issues should be addressed:

1) The abstract does not really report much about the findings of the work, but rather gives an introduction,. This should be changed to invoke interest.

The abstract was re-written to provide more details of the presented work.

2) RCD1 is referred to as an intrinsically disordered protein. I would call it a multi domain protein with IDRs.

Thank you for the suggestion, it is now changed in the text where appropriate.

3) Fig. 2B: Some NB localization retained?

Treatment with 3MB resulted in significant reduction of the NBs repeatedly in several experiments (here we show a representative image). However, it is possible that the inhibition of PARP activity in plants was not complete, thus, some NBs localization was still visible. Also, in mammalian field PAR is considered as a platform for liquid-liquid phase separation, and once established this platform is not required for maintenance of liquid-liquid phase separation.

4) Fig. 2C: When are GST and when are His-tagged proteins used? Why no anti-GST detection for GST-hWWE at left?

The GST fusion proteins were used in the dot blot assay for all proteins except for the full length RCD1, since RCD1-His protein was obtained with higher purity than GST-RCD1. The weaker signal from GST-hWWE with GST antibody on the right panel might be caused by lower protein amount. However, on the left panel the same GST-hWWE protein gave the same signal with anti-GST as other proteins.

5) Fig. 2D: Why are different recombinant proteins used? This makes control experiments needed (GST and His tags alone). Furthermore, protein is immobilized via amino-groups, which may affect both protein structure and interactions. This should be addressed.

RCD1-His protein was obtained with higher purity than GST-RCD1, therefore it was used in SPR analysis. Since GST alone did not show any signal on the dot blot assay, it was not analyzed with SPR. Moreover, if GST would bind PAR, it would be visible in case of the GST-RCD1 Δ WWE sensogram.

We also modified the text to explain why different recombinant proteins were used in the assays to make it clear for the readership.

6) Line 163: Here GST-RCD1 (and not His-RCD1) is used. Again what is used when and why?

GST-RCD1 was used in the *in-vitro* kinase assays since the protein purity is not that critical as for the label-free ligand-binding studies, and we used GST as a negative control protein. The GST-RCD1 is the major band in the corresponding Coomassie-stained images of the SDS-PAGE gels, and we can detect strong phosphorylation of this band by PPKs in autoradiographs.

7) Model presented in figure 4 is very interesting. However, it becomes very speculative when it comes to E3 ubiquitin-degradation. Could this be bioinformatically or experimentally addressed?

The question about proteasomal degradation of RCD1 was addressed experimentally by cell-free degradation assay using recombinant GST-RCD1 and protein extracts from Col-0 plants (new Supplementary figure 9). Protease inhibitors were added to the extracts before incubation with GST-RCD1. The experiment showed specific stabilization of RCD1 by the proteasomal inhibitor MG132. This is also addressed in the discussion.

8) Can the authors relate previous own work on the negative regulatory effects of RCD1 on DREB2 and NAC TFs more directly to this work?

It is addressed in the Discussion.

Reviewer #2 (Remarks to the Author):

PARylation is a reversible post-translational modification that modulates diverse physiological responses in eukaryotes. PARylation is mediated by poly(ADP-ribose) polymerase (PARPs). Plants have a group of unique PARP-like proteins, namely RCD1 and SROs. However, RCD1 and SROs do not have detectable PARP activities. This study shows that RCD1 functions as a PAR reader that directly binds PAR. RCD1 localizes to nuclear bodies in a PAR-dependent manner. The authors then identified PPKs as the kinases that phosphorylate RCD1 and regulate RCD1 abundance. This is the first report of a plant PAR reader, and elucidates its regulatory mechanism by protein phosphorylation. Together with a recent study about SRO2 possessing MARYlation activity that regulates protein stability, these studies expand the mechanistic understanding of plant-specific PARP-like protein functions. The manuscript was well written and easy to follow. The data are of high quality.

One point could be better explained. PPK-mediated phosphorylation reduces RCD1 abundance but is also required for RCD1 function. This seems to conflict.

Phosphorylation of the IDR2 modulates the protein function and stability. RCD1 abundance goes up when complemented with a non-phosphorylatable form yet this does not complement *rcd1* phenotypes, this suggests that phosphorylation is required for function and abundance. One way to interpret this is that RCD1 could be phosphorylated whilst bound to a transcription factor and phosphorylation mediated degradation follows on the complex, i.e. negative transcriptional regulation of gene expression.

In Fig. 3D, why GST-PPK was not labeled, and no phosphorylation was observed?

GST-PPK4 had higher activity towards the substrate (GST-RCD1), and the autophosphorylation band was not visible at this exposure.

Please also discuss whether phosphorylation mutant RCD1^{S/T}IDR2^A affects NB localization.

We addressed the question on the influence of RCD1^{S/T}IDR2^A mutation on NB localization experimentally by creating transgenic lines expressing RCD1^{S/T}IDR2^A-Venus. The new Figure 4B shows distinct NBs of different shape and number in case of RCD1^{S/T}IDR2^A-Venus compared to the NBs formed by wild type RCD1-Venus. It confirms an important role of IDR2 phosphorylation on RCD1 function, however, the nature and composition of these NBs requires further investigations.

Line 200, "The" should be "the".

It is now corrected.

Reviewer #3 (Remarks to the Author):

RCD1 is a disordered hub protein that has been previously shown to interact with over 30 TFs via its RST-domain. However, RCD1 also possesses other domains (WWE and PARP-like) that have been demonstrated to be functionally involved in facilitating PARylation and MARTylation within animal systems. Although no prior evidence has shown RCD1's involvement in these types of processes, the presence of these domains suggest that RCD1 may be involved in similar processes. Here, the authors excitingly propose that RCD1 possibly functions as a PAR reader by which RCD localizes to nuclear bodies by binding PAR via its WWE domain. Additionally, they show that PPKs co-localize in NBs with RCD1 to phosphorylate RCD1 and control its levels. Assuming the authors address the comments below, I recommend this paper for publication.

Major comments:

Does RCD1 actually interact with PARylated proteins? The authors use chemical analysis to show that PAR synthesis influences NB formation, However, in their Mass Spec analysis, do they find PARylated proteins that interact with RCD1 and if so, do these proteins localize to the same bodies?

The samples for mass spectrometry analyses were excised from SDS-PAGE gels (region corresponding to RCD1 band) either after immunoprecipitation of RCD1-HA with anti-HA antibody from plant extracts or from gels after *in-vitro* kinase reactions with recombinant proteins, where no poly(ADP-ribose)ylation occurs. Further the peptide mixture was enriched with TiO₂ for phosphopeptides; the phosphopeptides were then analyzed with CID fragmentation which is not favorable for ADP-ribosylated peptides due to their high lability (Bonfiglio JJ et al, 2017, Nucleic Acids Res.). So, in conclusion, detection of PARylated proteins was not technically possible due to the methods used.

Additionally, the test RCD1's ability to act as a PAR reader in vitro. However, this doesn't prove functionality as a PAR reader in vivo. Ideally, the authors should do a WWE domain swap and see if the plant and animal WWE domains are functionally interchangeable. I would suggest that if the authors cannot do this experiment, that they tone down their interpretation of RCD1 as a PAR reader.

We addressed the question on ability of RCD1 to bind PAR *in vivo* using co-immunoprecipitations with automodified PARP2-GFP. Proteins were extracted in buffer containing NAD and the auto-PARylated PARP2 was immunoprecipitated via the GFP tag. The result of the experiment is shown below: RCD1-RFP, but not RCD1ΔWWE-RFP, was immunoprecipitated with automodified PARP2-GFP (but not with GFP) (upper right panel on the figure below). This result suggests that the WWE-domain is required for the interaction.

Total extracts samples (left panels) were taken before immunoprecipitation to test for equal expression and loading. RFP-RNF146 and mutated RFP-RNF146 (RFP-RNF146 mut) were used as positive and negative controls. The expected bands are marked with asterisks on the RFP blots (upper panels).

Position of RCD1-RFP in PARP2-GFP pull-down (marked with yellow arrow, upper right panel) suggests formation of high molecular weight complex of yet unknown nature. We are a bit hesitant to include this data here – unless the editor would see that addition of this data, which does show the interaction *in vivo* would be beneficial. It shows the point asked, but in our view it is too preliminary to include before the composition and nature of the HMW complex is clarified, which is beyond the scope of this manuscript – and the results would merit a manuscript of their own.

We also modified the text to tone down the statements about RCD1 as a PAR reader *in-vivo*.

The authors use a transient assay to show that PPKs require RCD1 for their localization to nuclear speckles. However, since the authors have the *rcd1* mutant and fluorescently-tagged PPK1 proteins, they should confirm that PPKs require RCD1 for localization by expressing PPKs in the *rcd1* mutant background. Additionally, as the authors have the triple *ppk* mutants, they should test if PPKs are necessary for RCD1 localization. Instead the only elude to this using a transient system (*sup fig 5B*). Disordered proteins are concentration dependent and since transient assays often OX protein, it is hard to say if this is a meaningful result.

We addressed the question of PPK localization by transient expression of PPKs in *rcd1* mutant and by creating transgenic lines expressing PPKs in *rcd1* background. The results of these experiments are shown

below: in both cases, as expected, PPKs localized to NBs. The reason, according to our previous experiences, is the presence of SRO1 (paralog of RCD1 with similar domain structure) in *rcd1* mutant, which likely fulfills the role of the missing RCD1. RCD1 and SRO1 act in cells as both homo- and heterodimers, so when functional RCD1 protein is absent, the functional SRO1 dimer is still present. The double *rcd1 sro1* mutant is in practice not viable (Jaspers et al, 2009, Teotia & Lamb 2009), which, as the results suggest, prevents the experiment that would be required. However, the SRO protein family is conserved also in *N. benthamiana* and the results shown in the original submission already indicated that the speckle formation is species-specific, since expression of both Arabidopsis proteins was required for the speckle formation. The *N. benthamiana* SRO present in the nuclei was not able to recruit transiently expressed Arabidopsis PPKs to NBs. This is now also addressed in the revised text. Additionally, NBs can comprise of many constituent proteins, there may be other protein(s) involved in recruitment of RCD1 or the PPKs into the observed foci. The nature and composition of the NBs requires further investigations, which is beyond the scope of this manuscript and would produce a whole new manuscript.

Material: nuclei of mesophyll cells from 3-weeks old T1 plants

Transient expression of PPK1-RFP in Col-0 and *rcd1*

Minor comments:

Figure 1D: The authors propose that the NLS signal is important for the leaf curling phenotype displayed in the rcd mutants but it is not sufficiently captured in this image. Please provide a better image, in which the phenotype is sufficiently observed or downplay this result.

The *rcd1*-specific phenotype demonstrated in the Figure 1D is actually the leaf shape, not the leaf curliness. We are sorry for the misleading terminology; this is also corrected in the text. The rosette images were taken from 3-weeks old plants the same way as it has been done in other publications for easier direct comparison.

Supplemental 3A: The authors mention that 100µg of protein was added per lane but they do not have the necessary endogenous controls showing this is true. Please add an endogenous control.

The endogenous control (Rubisco large subunit, stained with amidoblack) is now added (Supplementary figure 3A).

minor editing suggestions:

Line 85: "trough" needs to be replaced with "through"

Line 86: and an "a" between "in" and "non"

All now corrected.

REVIEWERS' COMMENTS:

Reviewer #1 (Remarks to the Author):

I find that all important issues were addressed in the review process

Reviewer #3 (Remarks to the Author):

I want to thank the authors for their thoughtful experiments and rebuttals to all comments. I also agree that the PAR reader in vivo experiment is too preliminary, and am in support with how the authors decided to re-phrase the language. I support this article being accepted.